# INHABIT: A web-based decision support tool for invasive plant species habitat visualization and assessment across the contiguous United States

**Peder Engelstad**[1]*, **Catherine S. Jarnevich**[2], **Terri Hogan**[3], **Helen R. Sofaer**[2¤a], **Ian S. Pearse**[2], **Jennifer L. Sieracki**[3], **Neil Frakes**[4¤b], **Julia Sullivan**[5], **Nicholas E. Young**[1], **Janet S. Prevéy**[2], **Pairsa Belamaric**[5¤c], **Jillian LaRoe**[5]

**1** Natural Resource Ecology Laboratory, Colorado State University, Fort Collins, Colorado, United States of America, **2** U.S. Geological Survey, Fort Collins Science Center, Fort Collins, Colorado, United States of America, **3** National Park Service, Fort Collins, Colorado, United States of America, **4** National Park Service, Joshua Tree National Park, Twentynine Palms, California, United States of America, **5** Student Contractor to the U.S. Geological Survey, Fort Collins Science Center, Fort Collins, Colorado, United States of America

¤a Current address: U.S. Geological Survey, Pacific Island Ecosystems Research Center, Hawaiʻi National Park, Hawaiʻi, United States of America
¤b Current address: Bureau of Land Management, Ely, Nevada, United States of America
¤c Current address: Department of Forest and Wildlife Ecology, University of Wisconsin-Madison, Madison, WI, United States of America
* peder.engelstad@colostate.edu

**Data Availability Statement:** INHABIT is a publicly accessible website where users can view, copy,

## Abstract

Narrowing the communication and knowledge gap between producers and users of scientific data is a longstanding problem in ecological conservation and land management. Decision support tools (DSTs), including websites or interactive web applications, provide platforms that can help bridge this gap. DSTs can most effectively disseminate and translate research results when producers and users collaboratively and iteratively design content and features. One data resource seldom incorporated into DSTs are species distribution models (SDMs), which can produce spatial predictions of habitat suitability. Outputs from SDMs can inform management decisions, but their complexity and inaccessibility can limit their use by resource managers or policy makers. To overcome these limitations, we present the Invasive Species Habitat Tool (INHABIT), a novel, web-based DST built with R Shiny to display spatial predictions and tabular summaries of habitat suitability from SDMs for invasive plants across the contiguous United States. INHABIT provides actionable science to support the prevention and management of invasive species. Two case studies demonstrate the important role of end user feedback in confirming INHABIT's credibility, utility, and relevance.

## Introduction

A lingering problem in the complex interplay of conservation science and decision-making is how to resolve the disconnect between producers and users of scientific data. Digital

and download the presented data. In the variable sources tab, links to predictor layers are given whenever possible. Links to occurrence location databases and management polygons sources are provided in the F.A.Q. Additional resources for map data (and associated metadata) are available on ScienceBase at http://doi.org/10.5066/P92476V6.

**Funding:** Funding for this project was provided by the U.S. Geological Survey Invasive Species Program (https://www.usgs.gov/ecosystems/invasive-species-program), the U.S. Geological Survey Core Science Systems (CSS): Science Analytics and Synthesis (https://www.usgs.gov/core-science-systems/science-analytics-and-synthesis/about), and the U.S. Fish and Wildlife Service (http://www.fws.gov). The funders had no role in study design, data collection and analysis, decision to publish, or preparation of the manuscript.

**Competing interests:** The authors have declared that no competing interests exist.

communication platforms, (e.g., websites and web applications), are common solutions to bridge this gap and often fall under the umbrella of decision support tools (DSTs), software designed to fill knowledge gaps and support decision-making by analyzing and communicating information. Despite the ever-improving technological capabilities and functionality of DSTs, there continues to be a lack of uptake from practitioners, stakeholders, managers, and policy makers due to usability issues [1], misaligned policy priorities [2], lack of training or trust [3], and lack of relevance [4].

Surveys of end users highlight ways to improve DST adoption, like building long-term partnerships [4] and facilitating multidirectional dialogues between researchers, collaborators, and end users [5]. These efforts can help generate high-quality end user feedback and help frame decisions, especially in the early phases of the DST development cycle. These recommendations indicate that useful DSTs can arise within existing conservation science frameworks, such as translational ecology [6], which emphasize collaboration, co-ownership, and engagement. This method of co-producing data, via the synthesis of knowledge and experience from subject matter experts and the technical expertise of scientists, can lead to more actionable information and avoid landing off target from practical management-based needs.

Species distribution models (SDMs) can inform conservation and management decisions but a lack of communication between modelers and potential users continues to limit the integration of model outputs into decision-making [7]. Using statistical algorithms, SDMs can generate estimates of habitat suitability across space, can reveal drivers of species' distributions, and are an important tool for guiding management of both rare and invasive species. These models can inform sampling and surveillance design, and spatial prioritization decisions for management actions and regulatory decisions [8].

Although SDMs have important technical caveats and limitations, they are the primary tool for generating spatial predictions of habitat suitability for species presence. For invasive species, model outputs can quantify risk of establishment. Familiarity with SDMs can increase their uptake by managers and other decision-makers, but many people in these roles may lack the data, time, or expertise to develop their own models, even via relatively accessible platforms [e.g. 9, 10]. At the same time, model outputs associated with scientific publications are relatively difficult for decision-makers to independently discover and use [11]. Web-based DSTs have the potential to make SDM outputs more accessible, enhance efficiency by providing information on problematic invaders of interest to multiple user groups, and may be co-produced with users to deliver model results in an interpretable manner relevant to established decision-making processes.

Despite these benefits, we are aware of only a single attempt at this kind of DST, a regional effort by National Park Service (NPS) Inventory and Monitoring from 2008 that was only accessible offline. No known web-based DSTs with outputs from nationally scoped SDMs are readily available to invasive species managers in the United States. As a consequence, agencies that manage and plan for invasive species currently rely almost exclusively on occurrence location data from free, web-based aggregators such as the Early Detection and Distribution Mapping System (EDDMapS) [12], CalFlora [13], or the Global Biodiversity Information Facility (GBIF) [14]. These datasets describe historical and current known locations based on biased sampling efforts, lacking the full geographic coverage that suitability maps can provide. And while outputs from SDMs have been integrated with web-based DSTs in other areas of conservation science [15–19], many facets of data delivery leave room for improvement, including low spatial resolution maps, limited numbers of species, difficult to navigate interfaces, and irregular or inconsistent updates and maintenance.

There is a clear need to fill this information gap with a resource that provides high-resolution spatial predictions of invasive plant species habitat suitability, has ongoing maintenance

to meet shifting management use cases, and caters to a broad suite of potential end users. Therefore, we present a novel DST: the Invasive Species Habitat Tool (INHABIT; http://gis. usgs.gov/inhabit), a freely available resource designed to bridge the producer-user divide, drive knowledge sharing, and deliver credible SDM outputs to inform invasive plant species management at multiple spatial scales. This tool is a collaboratively and iteratively built R Shiny web application displaying spatial and aspatial summaries of ensembled SDMs covering the contiguous United States for numerous invasive plant species. All model outputs in INHABIT result from an established SDM methodology [20]. As a result, we focus on an introduction to the tool's design features, the collaborative nature of its development, and highlight case studies that demonstrate the impact and integration of user feedback to INHABIT's iterative evolution.

## Methods

### Initial design & development

The conceptual foundations leading to the creation of INHABIT emerged from conversations between NPS invasive species coordinators and U.S. Geological Survey (USGS) researchers around the need for, and communication of, SDMs for invasive species of concern. Management activities can be informed by SDMs at both local and regional scales, but model outputs need to be effectively distributed to land managers. Presentations of early web-platform mock-ups of INHABIT at workshops and conferences elicited feedback on desired content and design from early adopters including the NPS Invasive Plant Management Teams (IPMTs). These regional teams contribute to surveillance and control of invasive plant species, and restoration efforts at multiple parks within their regions. They also aid local managers through consultation and by providing training in invasive plant management techniques.

With input from IPMTs, NPS coordinators selected an initial set of species that had occurrence data, were regionally or nationally important, and were of high priority to potential early adopters. To aid in model development, IPMT representatives also provided feedback on the suite of potential predictors anticipated to be useful for modeling the distributions of these species. Coordinators from NPS and researchers from USGS expanded initial species lists with additional species of interest from other potential users, including the Bureau of Land Management and U.S. Fish and Wildlife Service. With multiple agency stakeholders, the platform needed to support management at multiple spatial scales within management units of differing sizes or across regions. The consensus between management stakeholders and researchers called for a platform that included: a relatively simple and intuitive user-interface for spatial and non-spatial data, compatibility with devices taken into the field, and the ability to rapidly provide user feedback.

A final consideration in the development of INHABIT was to deliver it via a platform that could regularly accommodate new iterations of SDM models and rapidly integrate improved interface features based on user community experiences (Fig 1). As early versions of INHABIT evolved, users actively engaged with the developers through presentations at scientific meetings attended by resource managers. Developers also presented to specific user groups to elicit feedback (e.g., Department of the Interior [DOI] invasive species community, Western Governors Association, internal agency invasive species groups). For example, USGS hosted a one-day workshop in 2019 with representatives from DOI agencies to further review INHABIT design and content. This workshop resulted in several changes to the tool including modifications to the map display, edits to the explanatory text to aid interpretation for non-modelers, and changes in statistical summary content. Workshop participants developed specific questions designed to improve user engagement with the inclusion of a feedback page.

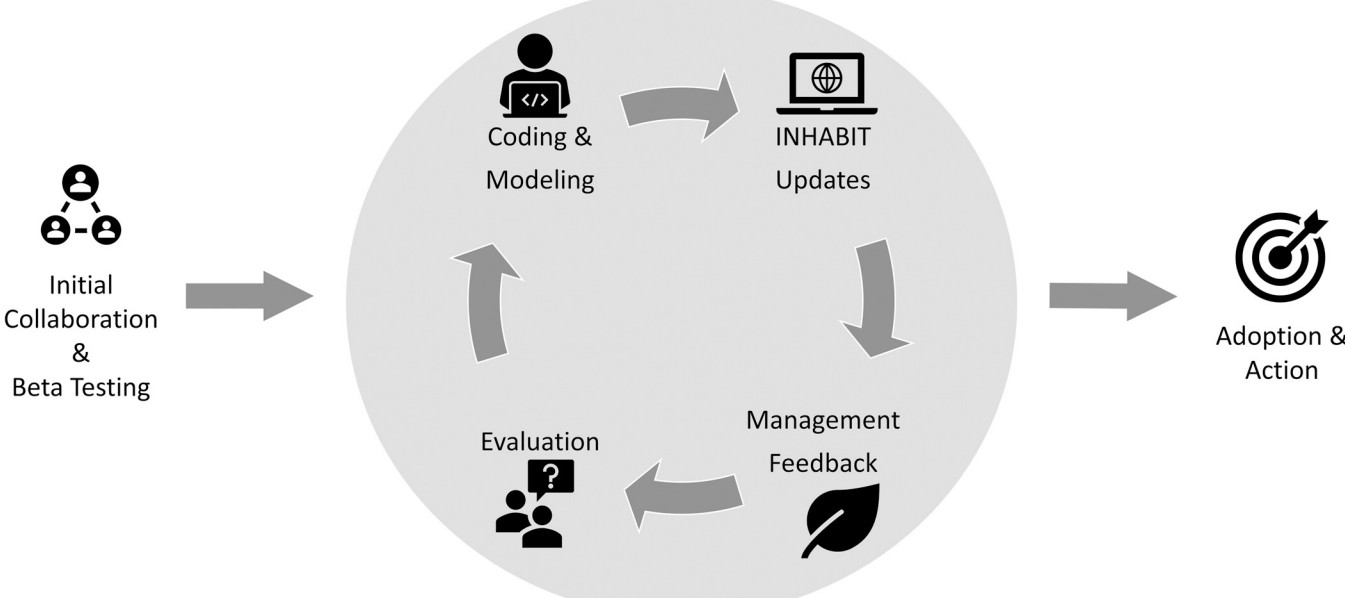

**Fig 1. A conceptual diagram illustrating the development of the Invasive Species Habitat Tool (INHABIT).** Co-production starts on the left and follows through to eventual adoption and land management or policy actions. The grey middle circle denotes the iterative cycle where updates are made to species distribution models and user interface elements.

## Architecture

For several reasons, we chose to deliver INHABIT's SDM map outputs and statistical summaries through the R Shiny web-application platform [21]. First, it is free, open-source software and is interoperable with all R libraries tested for this project. Second, Shiny apps are accessible simultaneously by multiple users, ensuring all users are working with the same version of the data. Third, Shiny's default stylesheet has a preconfigured responsive design, allowing the app to recognize different device types, an advantage when operating from differing physical locations (e.g., moving between office computers and field mobile devices). Finally, integrating changes to a Shiny app's codebase is simple and straightforward, which allows INHABIT to rapidly adapt to feedback from stakeholders through updates to the content and format of the displays. These features make it easy to address common challenges in DST development related to misinterpretation, load speeds, and the creation of meaningful functionality [22].

The data feeding INHABIT come from an established modeling workflow [20], wherein presence-background SDMs are fit using five statistical algorithms and two background point generation methods, yielding up to 10 total models per species. To build these models and generate spatial predictions of relative habitat suitability, the algorithms combine systematically vetted presence data (drawn from multiple online repositories) and species-specific spatial predictors (selected using natural history knowledge and drawn from a broader suite of predictors covering the contiguous United States). Each of the 10 models' mapped continuous predictions were subject to a set of four discrete thresholds to produce binary predictions. These binary predictions were summed to create ensembled maps (one per threshold) along a gradient of inclusivity, from a precautionary threshold predicting the greatest amount of suitable habitat to a targeted threshold focusing on areas with higher suitability values.

## Graphical user interface

Two main features comprise the graphical user interface of INHABIT: a left-hand sidebar consisting of user-defined inputs and a larger panel on the right with five navigational tabs (Fig 2). The sidebar is a static panel containing species and map-based selection toggles. The tabs atop the main panel allow users to navigate to different sections of the application: "Map", "Management Area Table", "Model Details", "About/F.A.Q.", and "Feedback". The "Map" tab shows spatial outputs: habitat suitability maps, management unit boundaries, and species occurrence locations. The "Management Area Table" displays information on the estimated relative risk of establishment posed by the focal species to each management area (e.g., national parks, wildlife refuges), including predicted amount of suitable habitat and the nearest known occurrence. The "Model Details" tab includes an array of technical information related to predictor variables, statistical model performance, and estimated species' responses to environmental drivers within the underlying SDMs. The "About" tab summarizes funding and data sources and answers frequently asked questions. The "Feedback" tab contains a form to provide comments and suggestions on website functionality and content or to suggest new species to add to the tool.

**Sidebar.** In the sidebar, users can select a species of interest by clicking into a drop-down menu to search by species' common name or scientific name. For each species, there are eight possible habitat suitability map visualizations, controlled by a slider bar. The maps are ensembles of up to 10 SDM outputs created using a continuum of four different threshold selection methods commonly used in SDMs to develop more or less targeted binary predictions across the landscape. They include: Minimum Predicted Presence (MPP), first percentile (0.01), tenth percentile (0.1), and Maximum Sensitivity Plus Specificity (MaxSS). The MPP threshold is the minimum predicted value for any of the occurrence points used in model generation while the first and tenth percentile threshold correspond to the value that would classify either 1% or 10% of the occurrence point locations used to fit the models as unsuitable. The MaxSS threshold optimizes the balance between model sensitivity and specificity, minimizing the mean of the error rate for both presence and background observations. Users can alternate between these four model thresholds using the slider bar based on their intended use and desire for a more precautionary (inclusive suitability) or targeted (more selective suitability) map. This qualitative range can help users select the map output most applicable to their

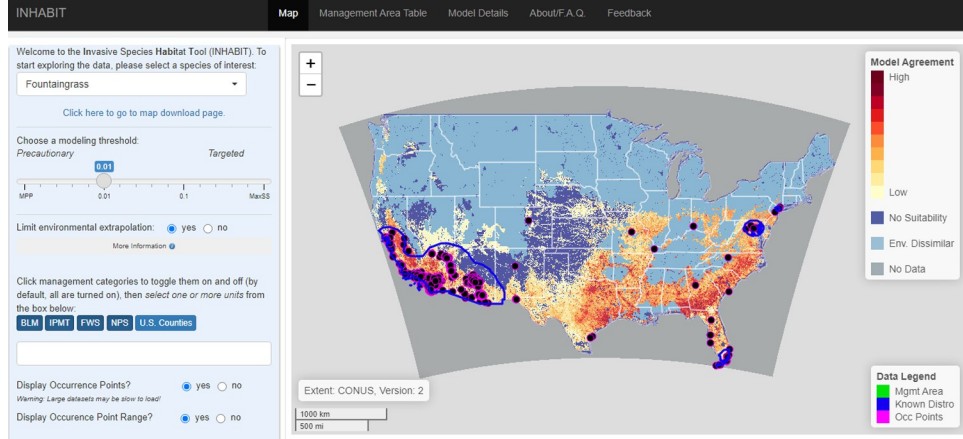

**Fig 2. The layout and features of the Invasive Species Habitat Tool (INHABIT; http://gis.usgs.gov/inhabit) landing page in August 2021.** The sidebar panel is visible on the left and the main panel on the top right with the "Map" tab selected.

specific management need (e.g., locating potential satellite populations). Based on stakeholder suggestions to reduce or clarify technical language, the slider bar range labels are qualitative ("Precautionary" to "Targeted") to better translate the meanings of these thresholds.

For all maps, users can also choose to mask areas where environmental conditions are outside the environmental range of occurrence points used to build the model by using the "Restrict Environmental Conditions" radio buttons. With the selection set to "Yes", environmentally dissimilar areas are displayed as light blue (Fig 2) rather than reflecting calculated ensembled suitability value for the selected threshold. Because model predictions are more credible within the environmental range of the training data, this feature addresses potential concerns about statistical extrapolation within the unrestricted map [23].

Management unit boundaries for multiple federal agencies and four counties are also searchable in the sidebar. Upon selection of a management unit, the map will zoom to that unit's extent or to the full extent of multiple selected unit boundaries. This feature enhances the utility of the tool for management and planning at the scale of individual management units (e.g., a focal park or refuge). Finally, users can also display the data used in model building, including the species occurrence locations and background sampling boundaries [20].

**Map tab.** By default, the main panel loads the "Map" tab (Fig 2), displaying the spatial predictions of ensembled SDMs. The map's color scheme represents the level of ensemble model agreement, with the darker colors indicating areas where more models predict suitable habitat and lighter colors where more models predict unsuitable habitat. This visualization choice provides a straightforward and quickly digestible way to distinguish spatial patterns on the landscape, giving users an initial 'gut check' on data credibility. As users modify the input parameters in the sidebar, the map will update accordingly based on species selection, model threshold, and occurrence location display options. Metadata about the extent considered in model development (contiguous United States or global for species with few U.S. records) and species model version are visible in the lower left-hand corner.

The habitat suitability data used to build the maps are an ensemble of up to 10 model outputs and cover the contiguous United States at a 90 m spatial resolution. Normally, maps this large can be notoriously slow to load with most R data visualization libraries. To address this, we used *gdal2tiles* (GDAL 2.4.1), a program that slices maps into thousands of smaller image files, which we subsequently upload to cloud-storage to ensure reasonable load times. This method also allows INHABIT to retain high spatial detail across zoom levels, providing the flexibility to accommodate users from local, regional, or national management contexts.

**Management area table tab.** The "Management Area Table" tab summarizes attributes of invasion risk within each management unit for the focal invasive plant species. The calculated values in the table are: "Estimated Suitable Area", "Percent of Unit Area", "Known Presence? (Count)", and "Minimum Distance to Known Occurrence". These data are quantitative, tabular complements to the map visualizations. Values in the table derive from the first percentile threshold of the habitat suitability map (with novel environments excluded) and are restricted to locations where a majority of the models in the ensemble indicate a location as 'suitable'. The first two quantitative columns in the table show the total number of acres of potential habitat within the boundaries of the management unit (defined as any location with an ensemble value greater than or equal to half the total number of models used), followed by the percent of the total unit area.

Especially for instances where the map may be more difficult to interpret, these numbers provide land managers with a quick assessment of overall risk posed by an invader. Tallies of the occurrence points used to develop the SDMs summarize the known state of infestation within a management unit ("Known Presence?") and the unit's proximity to known species locations ("Minimum Distance to Occurrence"). These columns can help determine the threat

status of an invasive plant species. Metaphorically, is it in the house, on the doorstep, or further down the road? Although the common language of scientific data is metric units, end user feedback indicated that managers would prefer imperial units. Summaries of habitat suitability and distance to the nearest known occurrence provide information on the potential extent of an invasion and its near-term likelihood. Taken together, these data can guide surveillance strategies at regional and local scales.

**Model details tab.** The "Model Details" tab is a resource for users wanting more technical information about the underlying models feeding INHABIT. Whereas the map conveys an aggregated version of the SDM output, the "Model Details" tab provides a more disassembled view of the individual models used within the SDMs to highlight the variation between model algorithms and background data selection methods. The "Model Details" tab contains

the nested sub-tabs of "Evaluation Metrics", "Variable Response Curves", "Variable Importance", "Variable Sources", and "Credibility Assessment" (Fig 3).

"Evaluation Metrics" displays a table of performance measures and accuracy statistics organized by model algorithm and background method. These include area under the receiver operating characteristic curve (AUC), area under the precision recall curve (AUC-PR), Correlation Coefficient, Percent Correctly Classified, Sensitivity,

Kappa, and True Skill Statistic. For detailed definitions of these metrics and the software used to compute them, see [10] for all except AUC-PR [24, 25]. These metrics provide multiple assessments of the performance of individual models, establishing transparency of the underlying model development process [8].

"Response Curves" presents a grid of graphs depicting the relationship between individual predictor values and relative habitat suitability, with a line for each model algorithm and background method retaining that predictor. The grid display orders by mean predictor relative importance across models, also represented with a numerical value in the upper left corner of each graph. (Fig 3B). Species experts can review environmental responses as they relate to the relative probability of occurrence and provide feedback indicating whether the response aligns with their knowledge of the species' biology and life history. Where the form of curves align with practitioners' expectations, these graphics can enhance the perceived credibility of model outputs and increase integration into decision-making frameworks. Where they do not, species experts can provide feedback for future iterations of models related to predictor inclusion.

"Variable Importance" provides a table of values indicating which environmental predictors are consistently influential in building the predictions across models (Fig 3C). As with the response curves, the output gives users an opportunity to compare estimated relationships to expert knowledge, including the ability to view the total number and relative ranking of predictors. "Predictor Sources" (Fig 3D) displays more detailed descriptions of each predictor used to model the selected species and provides hyperlinks to the original data or publications. Finally, "Credibility Assessment" (Fig 3E) includes an evolving portion of INHABIT: a template from [8] to interpret and assess the various aspects of SDM development process. In the future, this section will incorporate species-by-species information to reflect updates resulting from iterative model versions.

**About and feedback tabs.** The "About" tab provides a summary of INHABIT's development partnerships, funding sources, answers to frequently asked questions (generated through user feedback), and citation information. The "Feedback" tab provides a list of questions, many of which came from a one-day workshop that included the INHABIT team and representatives from DOI land management agencies. Many of the questions attempt to identify how users are currently implementing INHABIT in their management strategies, what additional species they would like added to INHABIT, and what aspects of the website's interface may be difficult to understand or use. Other questions attempt to solicit specific feedback to

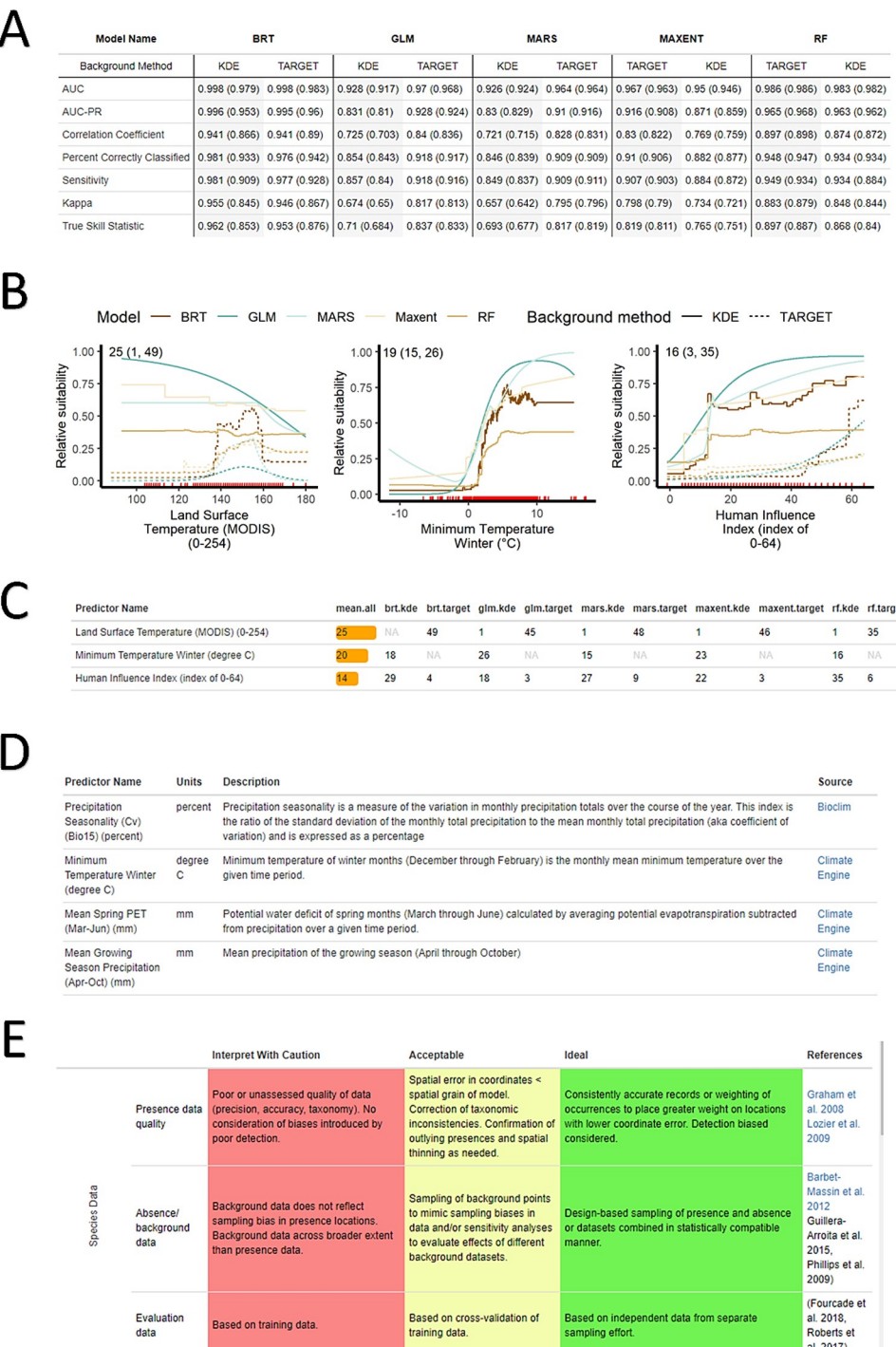

**Fig 3. Example content clipped from the "Model Details" tab.** Each chart's symbology also indicates model type: boosted regression trees (BRT), generalized linear model (GLM), multiadaptive regression splines (MARS), maximum entropy (MAXENT), random forest (RF), and background point generation method (target or kernel density estimate [KDE]). A) Evaluation metrics; in each column, the first number is the value with full model training data and the second, in parentheses, is the mean value from 10-fold cross-validation splits. B) Response curve images representing relative habitat suitability across environmental predictor values. Numbers in the upper left corner within each plot indicate mean, low, and high variable importance values reading from left to right. C) A tabular view of variable importance. Each row displays predictor names. D) Predictor descriptions and hyperlinks to variable sources, and E) the model credibility table based on the model assessment rubric from Tables 1a-c in [8]. Though it is a static template currently based on the overall modeling framework, the future version of this table will update its evaluative criteria relative to models for the selected species.

expand the credibility of INHABIT related to the SDM outputs, including evaluations of the ecological feasibility of the spatial predictors used or feedback about potential new occurrence location data for future model iteration. Finally, users can request new DST features, identify instructions or explanations that could be streamlined or more clearly explained, or reference missing capabilities found in other decision tools.

## Case studies

DST developers often have a good sense of what features are technically achievable or practical. But much of the fine-tuning and iteration comes from open dialogues around actionable user feedback and use cases. We acknowledge and attribute these suggestions on the "About" tab, helping remind visitors that INHABIT is a communal, collaborative effort. But the evolution of its relevance and utility is also the result of facilitated discussions with specific end users. The following case studies show how intensive engagement can help strengthen the bridge between scientists and practitioners and further demonstrate how INHABIT can inform ongoing management actions and complement existing planning frameworks.

### "Search and destroy" at Joshua Tree National Park

Individuals or small teams perform much of the work to control existing invasive plant species populations through activities such as backpack spraying, tractor mowing, and small-scale prescribed burns. Detection of new populations of invasive plants can be challenging given logistical, financial, and staffing constraints compounded by the time required to navigate rough or roadless terrain even in small management units. Spatial outputs from SDMs, like those found on INHABIT, can help guide sampling and survey efforts for known problematic invaders [26]. These efforts can also guide early detection and rapid response (EDRR) planning in regions without established invasive plant species populations [27].

One early adopter, the Lake Mead IPMT, saw potential for INHABIT to help with fountain grass (*Cenchrus setaceus* [Forssk.] Morrone), an invasive African bunch grass widely used as an ornamental plant. Fountain grass seeds are wind-dispersed and can travel great distances, making the species problematic in Joshua Tree National Park, a management unit located in the Southern California desert that collaborates with IPMT field crews to survey and treat fountain grass and other invasive plant species. The INHABIT team reached out to managers at the park to evaluate INHABIT and provide feedback on data pertaining to fountain grass, given the potential of the spatial outputs to complement ongoing treatment efforts by the IPMT in and around the park. The majority of known populations exist in the north and southwestern portions of the park, often adjacent to populated areas where the species has been planted. However, Joshua Tree National Park is an expansive and topographically diverse landscape, with large areas of the park only accessible by foot. New fountain grass infestations are usually found while navigating off-trail in wilderness settings, making searching for new satellite populations of fountain grass challenging.

Park staff indicated that accurate habitat suitability maps from INHABIT, combined with their local knowledge and expertise, help prioritize 'search and destroy' missions. Initial test drives of this process helped locate and treat remote fountain grass infestations (Fig 4), an independent validation of the predictions made by INHABIT models. This helped demonstrate the actionable nature of INHABIT's information and the effectiveness of the delivery mechanism for real-world management. Going forward, park staff at Joshua Tree National Park anticipate that INHABIT will help update invasive plant watchlists, which can be shared with volunteer groups to augment the survey capacity of park or IPMT field crews, especially in remote areas. Given the positive experience of these early adopters, INHABIT is now a part

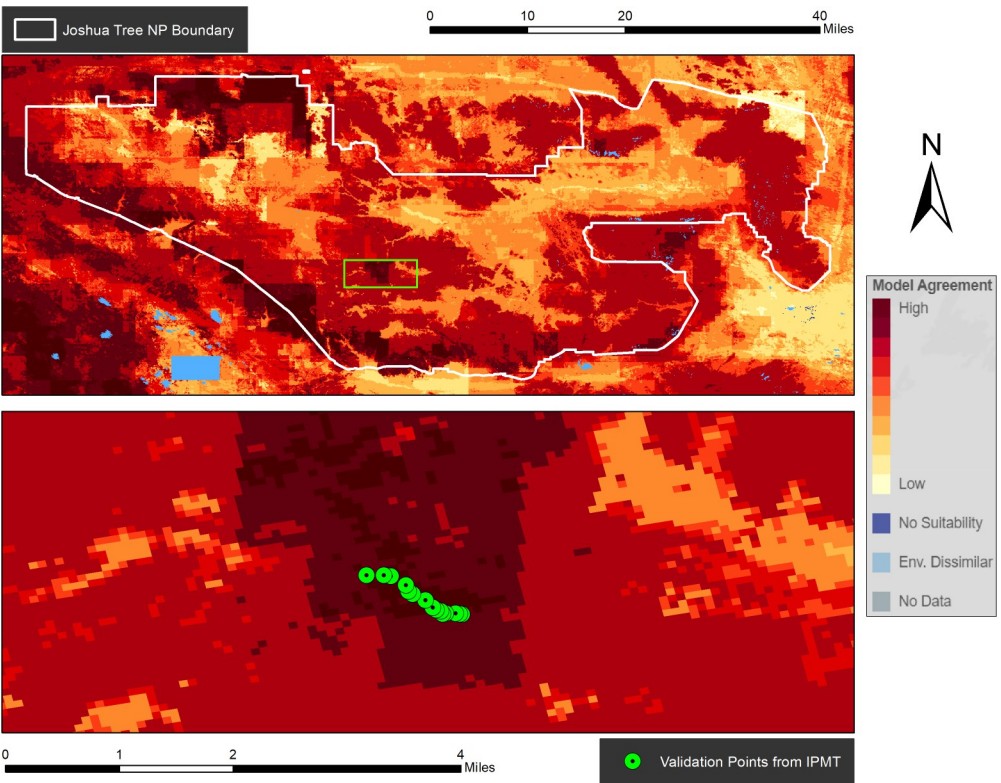

**Fig 4. Maps of fountain grass habitat suitability within Joshua Tree National Park.** The top map displays the full park extent while the bottom inset map displays the independently surveyed validation points (not included in the Invasive Species Habitat Tool [INHABIT] models) provided by the National Park Service.

of the NPS Pest & Invasive Species Management Project Kit, which provides step-by-step guidance for conducting invasive species and pest management at NPS units. The tool will be highlighted as a resource for individual park managers planning invasive plant prevention, EDRR, and control efforts.

## "Defined by water" with the U.S. Fish and Wildlife Service

Acknowledging and incorporating constructive feedback from end users is critical for the iterative improvement of DSTs. These interactions help ensure that end-products are relevant and meaningful to potential users, an important stepping stone to long-term adoption [4]. Additionally, collaborative design can help establish a community understanding of the standard for credible information between scientists and practitioners [28]. This understanding is important when the core products of a DST have the capacity to facilitate management recommendations. When parties can collaboratively improve the information and data that feed into DSTs, it becomes easier for all users to trust the validity and accuracy of downstream products.

As an example of this, the INHABIT team met with U.S. Fish and Wildlife Service (FWS) staff to discuss how the tool could integrate within the existing risk assessment and invasive plant prioritization frameworks used by their refuge system. From these conversations, FWS ecologists identified credibility issues with INHABIT's mapped predictions for species of concern within refuge boundaries. Their expert knowledge of these biological systems (e.g., wetlands and water-dominated landscapes) helped isolate where mapped predictions were

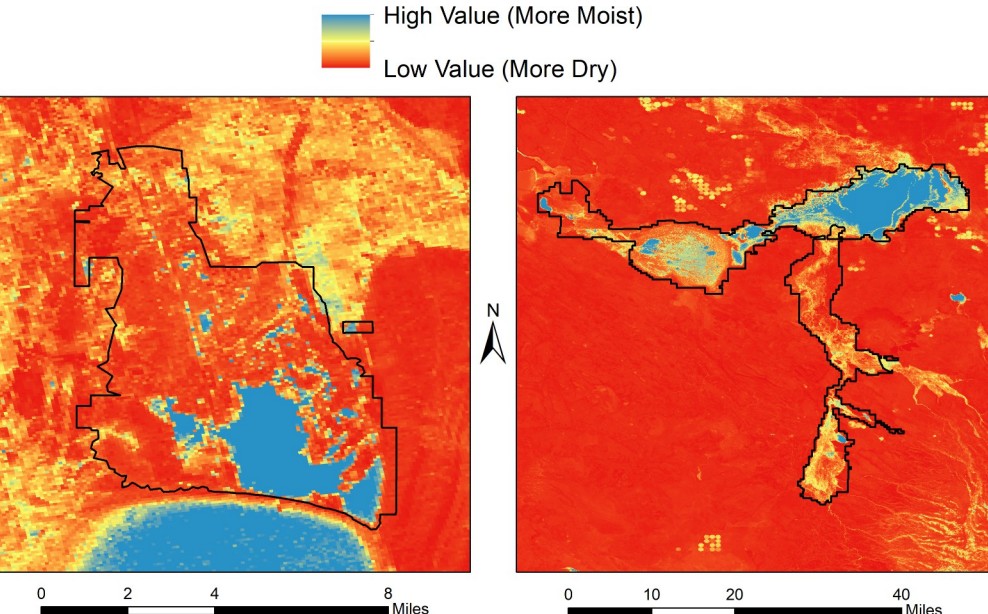

**Fig 5. Maps of median normalized difference moisture index (April to September).** This new predictor was reviewed favorably by the U.S. Fish and Wildlife Service for future model iterations due to its ability to delineate water bodies and seasonally inundated wetlands. Two example refuges from Oregon, USA are shown: Bear Lake Wildlife Refuge (left) and Malheur National Wildlife Refuge (right).

ecologically unreasonable and where the establishment of specific invasive plant species would be improbable.

Biologists from FWS described the refuge system as generally being "defined by water", emphasizing that the failure to appropriately capture wetland areas on the landscape could broadly undermine the utility of the tool for informing decisions on FWS units. Initially, INHABIT models lacked predictors to delineate and capture water and wetlands. In response, we built a suite of new predictors (S1 Table) including topographic indices indicating soil moisture potential, spectral time-series data identifying trends in water persistence, and high-resolution waterflow information from the National Hydrography Dataset. We shared these predictors and new model iterations with FWS biologists, who in turn evaluated which predictors were the most intuitive and plausible (Fig 5).

### Iterating INHABIT through feedback

The discussed use cases demonstrate the essential nature of ongoing conversations with current and potential users of INHABIT to continually add practical functionality and make the tool easier to use. One of INHABIT's strengths is its ability to rapidly address small user interface requests, giving regular users the confirmation that their voices are being heard. Iterative updates to the tool's visual style (e.g., map colors, scale bar, zoom limits, number of thresholds included), clarifications on technical language, and data additions (e.g., new management boundaries, new species, new predictors) have already been made as direct responses to user feedback.

Continued feedback and requests provide new and exciting ways in which INHABIT can grow and evolve. For example, conversations with FWS highlighted the utility of an alternate management summary option, which would summarize each species' habitat suitability maps and occurrence information within a focal management unit. The current table emphasizes

the relative risk posed by a single species to different management units, while the proposed table would compare risk across species for a single management unit. This 'ranked risk' approach for a selected management unit is an upcoming feature offering an alternative to the current management summary information, a suggestion whose utility has been echoed by other agencies, representing an entirely new use-case driven by ongoing user engagement and feedback. Integrating this feature into INHABIT directly addresses the type of quantitative support needed for multi-year planning efforts that FWS and other land management agencies develop within management units. These ranked risk summaries have broad support across agencies and address commonly understood management concerns like EDRR or watchlist generation.

Additional changes to INHABIT in active development also come from commonly received feedback. For example, to complement existing management datasets and allow INHABIT to more directly inform fieldwork, users can now download every map found on INHABIT [29]. Expanded selections for management boundaries will include National Forests from the United States Forest Service and we aim to develop partnerships with the Bureau of Indian Affairs to incorporate tribal lands and support tribal invasive species management. These and other planned enhancements (color scheme accessibility, expanded zoom levels, further simplification of map to three thresholds) will allow INHABIT to remain fresh and relevant as a unique resource in the management of invasive plant species.

## Conclusion

INHABIT is a web-based decision support tool designed to display actionable scientific data in the form of spatial and aspatial SDM outputs covering the contiguous United States in support of invasive plant species management. This tool is the outcome of a mutual vision by USGS researchers, NPS and other stakeholders to build a data sharing platform that will allow for rapid feedback from existing and new collaborators and regular updates to both models and user interface features. The two case studies highlighted here illustrate that expert knowledge of species-specific biology, frank discussions of credibility, and long-term partnerships are incredibly valuable when it comes to iterative development of both SDMs and user interface design of DSTs for conservation science. Iterative models can incorporate new occurrence data such as in case study one, new and updated predictors such as case study two (S1 Table), and other feedback related to the ecological plausibility of the model predictions.

Going forward, the flexible architecture of INHABIT's underlying R Shiny platform will allow it to evolve to meet the challenges of new data, unique use cases, or complex feature requests. Further down the road, integration with powerful cloud-computing resources could enhance the interactivity and flexibility of the tool, expanding its potential userbase. By remaining functionally agile, INHABIT can continue to facilitate actionable, science-based decision-making and improve the effectiveness and credibility of SDMs in the prevention of invasive plant species and prioritization of management decisions.

## Supporting information

**S1 Table. List of environmental variables considered for invasive habitat suitability modeling.** The variable column includes the name, listed within brackets are the unit of measurement, spatial (cell) resolution, and temporal resolution (if applicable). Each variable also has a specific description and source. Red variable names represent first version variables removed from the second version of the predictor list due to high correlations, redundancy, or contributing little importance to models. Yellow variable names represent second version variables that replaced a similar predictor from the first version. Green variable names represent second

version variables that were created in response to feedback and added to the second version of the predictor list to improve model sensitivity. Variable names without an associated color represent first version variables that were retained in the second version list without being modified.
(DOCX)

## Acknowledgments

We would like to thank the technical staff who worked tirelessly behind the scenes to bring this project to life: Kylie Baratta, Josh Beard, Paul Hoyeu, Tim Mancuso, Natalya Rapstine, Kristen Richardson, and Justin Wright. We also thank Curt Deuser, Bridgette Flanders, Jeremy Jirak, and Jess Wenick for beta testing and providing feedback for INHABIT. Any use of trade, firm, or product names is for descriptive purposes only and does not imply endorsement by the U.S. Government.

## Author Contributions

**Conceptualization:** Peder Engelstad, Catherine S. Jarnevich, Terri Hogan, Helen R. Sofaer, Ian S. Pearse, Jennifer L. Sieracki, Nicholas E. Young.

**Data curation:** Peder Engelstad, Julia Sullivan, Pairsa Belamaric, Jillian LaRoe.

**Formal analysis:** Peder Engelstad, Helen R. Sofaer, Ian S. Pearse, Jennifer L. Sieracki.

**Funding acquisition:** Catherine S. Jarnevich, Terri Hogan, Jennifer L. Sieracki.

**Investigation:** Peder Engelstad, Catherine S. Jarnevich, Helen R. Sofaer, Ian S. Pearse, Jennifer L. Sieracki, Neil Frakes, Julia Sullivan, Nicholas E. Young, Janet S. Prevéy, Pairsa Belamaric, Jillian LaRoe.

**Methodology:** Peder Engelstad, Catherine S. Jarnevich, Terri Hogan, Helen R. Sofaer, Ian S. Pearse, Neil Frakes, Nicholas E. Young, Janet S. Prevéy.

**Project administration:** Catherine S. Jarnevich, Helen R. Sofaer, Ian S. Pearse.

**Resources:** Terri Hogan, Jennifer L. Sieracki, Julia Sullivan, Jillian LaRoe.

**Software:** Peder Engelstad.

**Supervision:** Catherine S. Jarnevich, Helen R. Sofaer, Ian S. Pearse.

**Validation:** Peder Engelstad, Terri Hogan, Helen R. Sofaer, Jennifer L. Sieracki, Neil Frakes.

**Visualization:** Peder Engelstad.

**Writing – original draft:** Peder Engelstad, Catherine S. Jarnevich, Terri Hogan, Helen R. Sofaer, Ian S. Pearse, Jennifer L. Sieracki, Neil Frakes, Julia Sullivan, Nicholas E. Young, Janet S. Prevéy, Pairsa Belamaric, Jillian LaRoe.

**Writing – review & editing:** Peder Engelstad, Catherine S. Jarnevich, Terri Hogan, Helen R. Sofaer, Ian S. Pearse, Jennifer L. Sieracki, Neil Frakes, Julia Sullivan, Nicholas E. Young, Janet S. Prevéy, Pairsa Belamaric, Jillian LaRoe.

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
