## [Decision Letter · Decision Letter 0]

16 Aug 2021

PONE-D-21-14256

INHABIT: a web-based decision support tool for invasive species habitat visualization and assessment across the contiguous United States

PLOS ONE

Dear Dr. Engelstad,

Thank you for submitting your manuscript to PLOS ONE. After careful consideration, we feel that it has merit but does not fully meet PLOS ONE’s publication criteria as it currently stands. Therefore, we invite you to submit a revised version of the manuscript that addresses the points raised during the review process.

We look forward to receiving your revised manuscript.

Kind regards,

Daniel de Paiva Silva, Ph.D.

Academic Editor

PLOS ONE

1. Please ensure that your manuscript meets PLOS ONE's style requirements, including those for file naming. The PLOS ONE style templates can be found at https://journals.plos.org/plosone/s/file?id=wjVg/PLOSOne_formatting_sample_main_body.pdf and https://journals.plos.org/plosone/s/file?id=ba62/PLOSOne_formatting_sample_title_authors_affiliations.pdf.

2.We note that the grant information you provided in the ‘Funding Information’ and ‘Financial Disclosure’ sections do not match.

3. We note that Figures 2,4 and 5 in your submission contain [map/satellite] images which may be copyrighted. All PLOS content is published under the Creative Commons Attribution License (CC BY 4.0), which means that the manuscript, images, and Supporting Information files will be freely available online, and any third party is permitted to access, download, copy, distribute, and use these materials in any way, even commercially, with proper attribution. For these reasons, we cannot publish previously copyrighted maps or satellite images created using proprietary data, such as Google software (Google Maps, Street View, and Earth). For more information, see our copyright guidelines: http://journals.plos.org/plosone/s/licenses-and-copyright.

a. You may seek permission from the original copyright holder of Figures 2,4 and 5 to publish the content specifically under the CC BY 4.0 license. 

Additional Editor Comments (if provided):

Dear Engelstad et al,

In light of a new round of reviews, I believe your manuscript could be accepted for publication after you perform more improvements to it. I agree with reviewer #1 that more work is needed to make the MS fit into one of the two proposed categories, especially the category #1 indicated by reviewer #1.

Considering the necessary changes, I will provide you with a two-month period to improve your MS (resubmit by October 13th, 2021). In case you need more time, please let me know. By the time of re-submission do not forget to prepare a rebuttal letter where you inform your reviewers of all the accepted/rejected suggested with the related justifications.

Sincerely,

Daniel Silva, Ph.D.

Reviewers' comments:

Reviewer's Responses to Questions

**Comments to the Author**

1. Is the manuscript technically sound, and do the data support the conclusions?

Reviewer #1: Partly

Reviewer #2: Yes

2. Has the statistical analysis been performed appropriately and rigorously? 

Reviewer #1: N/A

Reviewer #2: Yes

3. Have the authors made all data underlying the findings in their manuscript fully available?

Reviewer #1: Yes

Reviewer #2: Yes

4. Is the manuscript presented in an intelligible fashion and written in standard English?

Reviewer #1: Yes

Reviewer #2: Yes

5. Review Comments to the Author

Reviewer #1: Thank you for this thoughtful article. The premise of the article is really important. I understood the premise as this: Researchers & scientists need to figure out who they are creating DST for, and engage those users early-on in the process of creating DSTs. I expected to read more details in the case studies about how the users interacted with INHABIT, how it was useful, what did and did not work for the users and why, and how you (the authors & DST developers) improved on the DST based on the users’ testing and feedback. I expected that all of this information from the case studies would be analyzed as data and thus be presented as a research paper. However, the article reads more like a description of the INHABIT DST. The data presented in the paper is related to the development of the invasive species habitat suitability models, and there is also a lot of detail on the functionality and design of the DST.

In my opinion, the information presented in this manuscript does not meet the PLOS ONE standards (https://journals.plos.org/plosone/s/reviewer-guidelines#loc-criteria-for-publication) for a research article. However, the INHABIT DST seems like a very useful tool with real-world applications, and I think that a paper describing the benefits of INHABIT, the assumptions of the model, and case studies should be published -- it just should not be labeled as a research paper based on the standards according to PLOS One. The manuscript does need major revisions.

I envision that you (the authors) can make edits and publish this one of two ways.

1) A manuscript that falls within the category of PLOS Ones’ “Methods, software, databases, and tools” paper:

https://journals.plos.org/plosone/s/submission-guidelines#loc-methods-software-databases-and-tools. The manuscript is currently much more like this type of an article than a standard PLOS ONE research article. I don’t think that you need to make as many edits in order to conform to PLOS ONE standards for this kind of submission.

Based on the guidelines at the URL above, this manuscript could be edited to focus MORE on the following aspects of INHABIT & the two case studies (the two sections in quotes below are copied from the URL above):

Utility

“The tool must be of use to the community and must present a proven advantage over existing alternatives, where applicable.…For software, databases, and online tools, the long-term utility should also be discussed, as relevant. This discussion may include maintenance, the potential for future growth, and the stability of the hosting, as applicable.”

- provide more details on the proven advantage over existing alternatives

- potential for future growth? Perhaps there are insights from the case studies

- the long term utility is implicit, but again, give more insights from the case studies, for example, any surveys from the users, or contracts with other users, or feedback submitted to the platform?

Validation

Submissions presenting methods, software, databases, or tools must demonstrate that the new tool achieves its intended purpose. If similar options already exist, the submitted manuscript must demonstrate that the new tool is an improvement over existing options in some way. This requirement may be met by including a proof-of-principle experiment or analysis; if this is not possible, a discussion of the possible applications and some preliminary analysis may be sufficient.

- the case studies do show validation

or,

2) With a lot more editing, this could be a standard research article that includes a lot more data (qualitative and quantitative, if available, plus any statistical analyses?) on the case studies, e.g., the feedback from participants in both groups, how you elicited their feedback, sample sizes, how you iterated on the design of the DST and/or co-created with the users your current version of INHABIT. What did you learn from the users and what kinds of questions elicited the learnings? How do you know that the user groups will continue to benefit from INHABIT? There seems to be a lot of great data from the FWS case study. It’s really great that you created new predictors to meet their stated needs, and that FWS was generally pleased by this—but these learnings are really glossed over it in lines 318-352. Also, what kinds of interesting feedback do you get submitted via the feedback form? How can this help further develop the DST? In order to present this as a research article that is more about user testing & iteration, perhaps look to some of the methods that entrepreneurs use when starting new businesses (see for example Lean Startup methodology).

Reviewer #2: Paper reviewed: “INHABIT: A web-based decision support tool

for invasive species habitat visualization and assessment

across the contiguous United States” by P. Engelstad, C.S. Jarnevich, T. Hogan, H.R. Sofaer, I.S. Pearse, J.L. Sieracki, N. Frakes, J. Sullivan, N.E. Young, J.S. Prevéy, P. Belamaric and J. LaRoe

Reviewer: Prof. V.K. Tokhtar, Doctor of Science (Biology),

Belgorod State National Research University (Russia)

Preserving biodiversity is among the main challenges faced by contemporary science. Solution to this problem may only be found in the course of cross-disciplinary research. Good prospects are offered by the use of mathematical approaches in natural sciences. The latter is in the focus of the paper reviewed.

The title and abstract are appropriate for the context. The paper is well-constructed, and the findings are sufficiently described.

The authors provide the results of developing the Invasive Species Habitat Tool (INHABIT), a web application DST built with R-Shiny to display spatially predicted habitat suitability from SDMs for invasive plants across the continental United States. The most important thing is that INHABIT provides actionable science to support the prevention and management of invasive species. Of equal importance is the fact that INHABIT is a publicly accessible website where users can view, copy, and download the presented data.

The authors describe the web-based decision support tool allowing for visualization of invasion species habitats across the contiguous United States. The tool bears testimony to the reliability of user data and facilitates research. INHABIT makes it possible to track the spread of invasive species, for instance, Cenchrus setaceus [Forssk.] Morrone (as exemplified with the case of Joshua Tree National Park) with a view to preventing its dispersal. A substantial advantage of the paper is the abundance of relevant graphic materials.

I believe the paper “INHABIT: A web-based decision support tool for invasive species habitat visualization and assessment across the contiguous United States” definitely deserves scientific attention, and the approaches described by the authors should be further implemented. I therefore recommend the paper for publication in PLOS ONE journal.

6. PLOS authors have the option to publish the peer review history of their article (what does this mean?). If published, this will include your full peer review and any attached files.

Reviewer #1: No

Reviewer #2: **Yes: **Valeriy K. Tokhtar

---

## [Author Response · Author response to Decision Letter 0]

12 Nov 2021

Dr. Silva,

Thank you for giving us the chance to submit a revised version of our manuscript. We are pleased that the reviewers valued the contribution provided by our manuscript, and appreciate the thoughtful comments and suggestions you and the reviewers were able to provide. As suggested by Reviewer #1, our manuscript now reflects important changes to more closely align with PLOS ONE’s criteria for “Methods, software, databases, and tools”. Namely, we have reworked the introduction to provide more detail on the methods currently used to generate invasive plant management actions and include more detail on the novelty of the tool in this context. We also now include a third subsection to the case studies that expands upon the ways in which INHABIT has and will change as a direct result of user feedback.

In addition to the above comments, figure two has been revised to remove any copyright content. Figures four and five were flagged for possible copyright violations but they both only contain image content from our model outputs. Minor comments have been incorporated, and we have edited for clarity where suggested. See below for detailed responses to both reviewers.

Comments for Reviewer #1 

We appreciate the reviewer’s suggestion that, as a description of a real-world application, the content of this manuscript falls under the “Methods, software, databases, and tools” category. To better structure the manuscript to fit the criteria of these types of submissions (namely, Utility and Validation), the reviewer suggested the following:

“provide more details on the proven advantage over existing alternatives “

We have edited a significant portion of the introduction to emphasize the novelty of this tool and describe its difference from currently available tools. From conversations with land managers and practitioners, we are not aware of any available tool equivalent to INHABIT. Without INHABIT, occurrence locations alone shape much of the spatial aspect of invasive species management and assessment in the United States. The use of occurrence locations in isolation paints an incomplete picture of the landscape and so can be complemented and filled in by predictions of habitat suitability from species distribution models. Overall, our tool represents a fundamental shift in the accessibility, interpretability, and utility of species distribution model outputs for the explicit purpose of invasive species management.

“potential for future growth? Perhaps there are insights from the case studies”

“the long term utility is implicit, but again, give more insights from the case studies, for example, any surveys from the users, or contracts with other users, or feedback submitted to the platform?”

To address these closely related comments, we have added a third section to the case studies entitled “Iterating INHABIT through feedback”. In this new section, we go into detail regarding the changes (big and small) to INHABIT that have already been implemented as a direct result of user feedback. Also highlighted are upcoming changes and their anticipated impact on the invasive species management community at large. The changes to the website come from a variety of sources, including the feedback tab, ongoing partner conversations, and web analytics analysis. In the future, a more formal usability survey may be conducted to help determine specific enhancements to web-design elements of INHABIT.

“the case studies do show validation”

We agree with the reviewer that validation is an important part of our work. Collaboration from the National Park Service has been very fruitful in providing on-the-ground confirmation of the SDM predictions available through INHABIT. Continually evolving validation strategies are also anticipated for future model workflow iterations, with more robust testing and training splits building off the current methods from Young et al, 2020.

Comments for Reviewer #2

We thank the reviewer for their enthusiasm about our work and agree that INHABIT is a boon to management planning through ease of access, downloadable materials, and unique visualizations. We appreciate the reviewer’s recommendation for publication and that, while the edits mentioned above represent changes to the manuscript, the fundamental structure has not been altered significantly. 

We look forward to hearing from you in due time regarding our submission and to respond to

any further questions and comments you may have.

Sincerely,

Peder Engelstad

peder.engelstad@colostate.edu

+1 206-478-1474

9/21/2021

---

## [Decision Letter · Decision Letter 1]

12 Jan 2022

INHABIT: a web-based decision support tool for invasive plant species habitat visualization and assessment across the contiguous United States

PONE-D-21-14256R1

Dear Dr. Engelstad,

We’re pleased to inform you that your manuscript has been judged scientifically suitable for publication and will be formally accepted for publication once it meets all outstanding technical requirements.

Kind regards,

Daniel de Paiva Silva, Ph.D.

Academic Editor

PLOS ONE

Additional Editor Comments (optional):

Dear Engelstad et al.,

It is with a great pleasure that I inform you that your manuscript was accepted for publication in PLoS One!

Good job!

Sincerely,

Daniel Silva

Reviewers' comments:

Reviewer's Responses to Questions

**Comments to the Author**

1. If the authors have adequately addressed your comments raised in a previous round of review and you feel that this manuscript is now acceptable for publication, you may indicate that here to bypass the “Comments to the Author” section, enter your conflict of interest statement in the “Confidential to Editor” section, and submit your "Accept" recommendation.

Reviewer #1: All comments have been addressed

Reviewer #2: All comments have been addressed

2. Is the manuscript technically sound, and do the data support the conclusions?

Reviewer #1: Yes

Reviewer #2: Yes

3. Has the statistical analysis been performed appropriately and rigorously? 

Reviewer #1: N/A

Reviewer #2: Yes

4. Have the authors made all data underlying the findings in their manuscript fully available?

Reviewer #1: Yes

Reviewer #2: Yes

5. Is the manuscript presented in an intelligible fashion and written in standard English?

Reviewer #1: Yes

Reviewer #2: Yes

6. Review Comments to the Author

Reviewer #1: Dear Peder,

Thank you for addressing my initial comments on the format of the paper so that it better fits Plos’ requirements of the methods, tools, etc. manuscript. This version is much improved! I have a much better understanding of how stakeholders can use INHABIT and how easy it is to adapt the tool to the needs and on-the-ground knowledge of users. Lines 377-387 is exactly what I was looking for in my previous review. Thank you for explaining a type of feedback, how you are adapting INHABIT, and why the change is useful.

I only have a few minor editorial suggestions to improve the clarity of sentences in a few locations.

I hope that this publication helps you reach more additional users. In addition, and not something that needs to be included in the publication, I hope that you’re considering training sessions and/or scheduling sessions at relevant scientific meetings to recruit additional users & continue to get feedback.

Minor suggestions:

Line 146:

Rather than stating that the modeling workflow is “…established by [citation 20]…”, can you specify the study’s authors and/or project in narrative form in the sentence?

Lines 364-367 (Figure 5 description)

Use of the word "positively evaluated" here is a little confusing. I think you are saying they (USFWS) “validated” the new predictor? Or that they evaluated the modifications, and agreed that the updates were more accurate/precise/realistic? Please consider rewording this to better reflect USFWS’s reaction.

Line 388: This sentence is a bit awkward -- I think the sentence is supposed to set up the rest of the paragraph to provide examples of commonly received feedback? If so, please consider rewording it for clarity so that readers know they are about to read about examples.

Lines 401-403: I suggest breaking up this sentence into >1 sentence for clarity. Specifically, the last clause in the sentence – it’s a bit unclear who or what can “…provide regular iteration…”.

Lines 406-407: This is the first place you mention case study 1 and 2 in the narrative, and while I assume case study 1 is “Search and Destroy” at Joshua Tree, it would be good to update the headings for the case study sections.

Reviewer #2: Conduct similar studies in other National Parks (in other states), expand the list of sites and compare the results

7. PLOS authors have the option to publish the peer review history of their article (what does this mean?). If published, this will include your full peer review and any attached files.

Reviewer #1: No

Reviewer #2: **Yes: **Valery Tokhtar, Dr. Sci (Biol)

---

## [Editor Report · Acceptance letter]

31 Jan 2022

PONE-D-21-14256R1 

INHABIT: a web-based decision support tool for invasive plant species habitat visualization and assessment across the contiguous United States 

Dear Dr. Engelstad:

I'm pleased to inform you that your manuscript has been deemed suitable for publication in PLOS ONE. Congratulations! Your manuscript is now with our production department. 

Kind regards, 

on behalf of

Dr. Daniel de Paiva Silva 

Academic Editor

PLOS ONE